# Antibacterial and anti-virulence effects of *Saxifraga stolonifera* Meeb extracts against *Pseudomonas aeruginosa*

Weidong Chen,[1] Zijie Zhang,[2] Yuanchun Huang,[3] Lin Chen,[4] Yijing Zhuang,[5] Yue Li,[6] Yuxiang Hong,[7] Lei Liu,[8] Qin He,[2] Qing Peng,[9] Fen Yao[2]

**ABSTRACT** *Saxifraga stolonifera* Meeb is widely used as a traditional Chinese medicine for the treatment of infections. This study aims to evaluate the antibacterial properties and suppression of virulence by *Saxifraga stolonifera* Meeb extracts on *Pseudomonas aeruginosa*. Following extraction of *Saxifraga stolonifera* Meeb with petroleum ether, ethyl acetate, n-butyl alcohol, and water, the n-butyl alcohol extract had the strongest activity against *P. aeruginosa* PAO1 and *P. aeruginosa* ATCC27853, with minimum inhibitory concentration (MIC) values of 10 and 5 mg/mL, respectively. In the presence of the n-butyl alcohol (n-BuOH) extract at 1/4MIC, genes *lasI*, *lasR*, *rhlI*, *phzA1*, *phzA2*, and *pilG* were decreased to levels ranging from 13% (*lasI*) to 43% (*phzA2*). Both biofilm formation and pyocyanin production of PAO1 were inhibited by the n-BuOH extract at sub-inhibitory concentrations. N-butyl alcohol extract analyzed by HPLC-Q-TOF-MS/MS showed more than 11 compounds. Overall, our results suggest that the n-BuOH extract from *Saxifraga stolonifera* Meeb may be used as a new anti-virulence agent for *P. aeruginosa* infection.

**IMPORTANCE** *Pseudomonas aeruginosa* infections pose severe challenges to clinical treatment, and anti-virulence therapy has emerged as a novel therapeutic strategy. This study demonstrates that the n-butanol extract of *Saxifraga stolonifera* exerts anti-virulence effects by downregulating virulence-related genes, inhibiting quorum-sensing systems, and biofilm formation. Moreover, its multiple bioactive components also possess antibacterial and anti-virulence properties. *S. stolonifera* is thus promising to be developed into a novel anti-virulence inhibitor against *P. aeruginosa* for the prevention and treatment of clinically relevant infections.

**KEYWORDS** quorum-sensing, biofilm, pyocyanin, HPLC-Q-TOF-MS/MS

*P*seudomonas aeruginosa is an important gram-negative opportunistic pathogen that causes soft tissue, urinary tract, and chronic pulmonary infections (1). As an important pathogen, *P. aeruginosa* infection is ultimately responsible for morbidity and mortality in patients with cystic fibrosis (CF)(2, 3). Treatment has become increasingly challenging due to the dramatic increase in antibiotic resistance of *P. aeruginosa* (4).

As a potential approach for new antibiotics, targeting virulence factors has gained increasing attention (5). First, by decreasing the expression or activity of virulence factors, the ability of bacteria to colonize and invade the host is reduced. In addition, in contrast to traditional antibiotics, suppression of virulence does not directly kill the bacteria, leaving the bacteria with less pressure to evolve into resistant clones. Furthermore, this inhibition does not interfere with the normal microbial flora and may also allow the host immune system to prevent bacterial colonization or clear any established infection (6, 7). Quorum-sensing (QS), a process of cell-to-cell communication modulating population density-dependent group behavior and controlling the expression of virulence genes,

**Peer Reviewers** Iqbal Ahmad, Aligarh Muslim University, Aligarh, India; Biao Tang, University of the Chinese Academy of Sciences, Hangzhou, China

Address correspondence to Fen Yao, fyao@stu.edu.cn, or Qing Peng, pengqing@cuhk.edu.cn.

Weidong Chen and Zijie Zhang contributed equally to this article. The author order was determined by the seniority order.

The authors declare no conflict of interest.

See the funding table on p. 11.

has been extensively studied (8), and could provide new targets for screening new antimicrobial agents. There are four QS systems in *P. aeruginosa*, namely the *las*, *rhl*, *pqs,* and *iqs* systems (8, 9). The transcriptional regulator *lasR* and the cognate autoinducer synthesized by *lasI* constitute the *las* system, while *rhlI* and transcriptional regulator *rhlR* constitute the *rhl* system (10). Some natural anti-virulence agents that inhibit QS systems have also been identified (11, 12).

*Saxifraga stolonifera* Meeb is a traditional Chinese herb that can be found in most areas of southern China. It has been used for hundreds of years for the treatment of benign prostatic hyperplasia (BPH), otitis media, respiratory disease, and skin and wound infections in China (13, 14). However, activity against bacteria has yet to be confirmed. In particular, whether the extracts from *S. stolonifera* Meeb have anti-virulence effects against *P. aeruginosa* still remains unclear. In preliminary screening with selected Chinese medical herbs, we found that ethanol extracts from *S. stolonifera* Meeb had good activity against *P. aeruginosa*, with MICs ranging from 25 to 50 mg/mL. This suggests that *S. stolonifera* Meeb extracts might be a promising reagent used for *P. aeruginosa* infection and could also be used as herbal feed additives to address the increasingly severe issue of antibiotic resistance in animal husbandry (15, 16). In this study, we characterize the anti-*P. aeruginosa* effect of different fractions of an ethanol extract from *S. stolonifera* Meeb. Furthermore, the inhibitory effects of *S. stolonifera* Meeb extract on virulence gene regulatory systems, such as the QS system and virulence factors, were also investigated.

## MATERIALS AND METHODS

### Plant material and chemicals

Fresh leaves and stalks of *S. stolonifera* Meeb were collected from a rural area in Puning city, Guangdong province, China. The stalks and leaves were dried in an oven at 50°C, then ground to fine powder with a blender. Gallic acid, protocatechuic acid, bergenin, and chlorogenic acid were purchased from Shanghai Yuanye Bio-Technology Company Limited, China.

### Extraction

The leaves and stems of *S. stolonifera* (93 g) were extracted with 95% EtOH (3 × 1 L, every 2 h, 40°C) under ultrasonication. The solution was filtered, and the collection was evaporated in vacuum to obtain a residue (9.2 g). The residue was suspended in $H_2O$ (0.5 L), then successively extracted at room temperature with 4 × 0.5 L of petroleum ether, EtOAc, and n-butyl alcohol (n-BuOH). Each solution was concentrated under reduced pressure (at 50°C) to result in a petroleum ether extract (1.947 g), ethyl acetate extract (1.582 g), n-BuOH (1.056 g), and water extract (4.615 g).

### Determination of MICs

To compare the antibacterial activity against *P. aeruginosa in vitro*, MIC values were determined using broth microdilution method according to the Clinical and Laboratory Standard Institute guidelines (17). Each extract was thoroughly mixed with 20% methanol as the stock solution and diluted with sterile water. Gallic acid, protocatechuic acid, bergenin, and chlorogenic acid were dissolved in sterile water. Briefly, a 50-μL volume of each solution was dispensed into each well of a microtiter plate, followed by 50 μL Mueller-Hinton Broth containing $10^6$ CFU (colony-forming units)/mL of *P. aeruginosa*. The plates were incubated at 37°C for 24 h. The growth of microorganisms was observed as turbidity at 600 nm, being measured with a SpectraMax M2e (USA). The MIC was determined as the lowest concentration that completely prevented microbial growth. In addition, the MICs of ceftazidime against PAO1 and ATCC 27853 were also determined as a reference. All samples were tested in triplicate.

## Bacterial strains and media

The bacterial strains used in this study are shown in Table 1. *P. aeruginosa* PAO1 and ATCC 27853 strains were grown and maintained in Luria-Bertani (LB) broth, Mueller-Hinton (MH) broth, or on LB or MH agar. Chromosomal fusion lux-based reporter strains were the gifts from Dr. Kangmin Duan (18).

## Detection of virulence gene expression by luminescence assay

The sub-inhibitory concentration that did not inhibit the growth of PAO1 was determined by measuring the growth curve, using a SpectraMax M2e (USA). The expression of genes was determined using the lux-based reporters: the promoter regions of the gene were cloned into the upstream of the lux genes on plasmid pMS402, gene expression in liquid cultures was determined as counts per second (cps) of light production, with a Molecular Devices FilterMax F5 (USA) (18). An overnight culture of a PAO1 lux reporter was diluted 1:100 in MH broth and incubated at 37°C for 3 h. Each culture was then adjusted to $OD_{600}$ = 0.3. In each well of a black, clear-bottom 96-well plate (Costar 3614), 10 µL PAO1 lux reporter culture was inoculated into 90 µL MH broth containing a sub-inhibitory concentration of extract/compounds. The same volume of PAO1 culture inoculated in MH broth without drug was set as the control. Luminescence was measured and the plate was shaken every 30 min for 24 h, using the microplate reader (Filter Max F5, USA). Each assay was repeated three times separately.

## Biofilm formation and pyocyanin production

Quantitative analysis of biofilm formation was performed, in a 96-well polystyrene plate, according to the method of Kessler et al with some modifications (19). An overnight culture of PAO1 was diluted to $OD_{600}$ =0.3. Then 50 µL bacterial culture was added into a 96-well plate, followed by addition of 50 µL MH culture medium, with or without n-BuOH extract and four identified compounds (gallic acid, protocatechuic acid, bergenin, and chlorogenic acid) at sub-inhibitory concentrations. The plates were incubated at 37°C as static cultures for 24 h. Following incubation, 25 µL 1% crystal violet stain was added to each well at room temperature. Fifteen minutes later, the liquid was removed, and the wells were washed four times with water. The remaining stained biofilm was dissolved in 200 µL 95% ethanol for 10 min with rocking. Then, a 150-µL aliquot was removed and transferred to another 96-well plate, and the optical density of the dissolved stain was measured at $OD_{600}$. Microtiter wells containing medium and extract were used as blanks. Each assay was repeated three times.

The pyocyanin production assay is based on the absorbance of pyocyanin at 520 nm in acidic solution (20). PAO1 was grown in LB broth, with or without n-BuOH extract and the four identified compounds at sub-inhibitory concentrations, at 37°C for 3 days. One-milliliter supernatant of each culture was removed and then extracted with 0.6 mL of chloroform, then re-extracted with 0.2 mL of 0.2 N HCl. The absorbance of this solution was measured at 520 nm by use of the microplate reader (SpectraMax M2e, USA).

TABLE 1 Bacterial strains used in this study[a]

| Strain | Description | Source or reference |
|---|---|---|
| *PAO1* | Wild-type of *P. aeruginosa* | This laboratory |
| *ATCC27853* | American Type Culture Collection | This laboratory |
| *PAO1 phzA1-lux* | *phzA1-luxCDABE* genomic reporter fusion in PAO1, Tc resistant | (18) |
| *PAO1 phzA2-lux* | *phzA2-luxCDABE* genomic reporter fusion in PAO1, Tc resistant | (18) |
| *PAO1 rhlR-lux* | *rhlR-luxCDABE* genomic reporter fusion in PAO1, Tc resistant | (18) |
| *PAO1 rhlI-lux* | *rhlI-luxCDABE* genomic reporter fusion in PAO1, Tc resistant | (18) |
| *PAO1 pilG-lux* | *pilG-luxCDABE* genomic reporter fusion in PAO1, Tc resistant | (18) |
| *PAO1 lasI-lux* | *lasI-luxCDABE* genomic reporter fusion in PAO1, Tc resistant | (18) |
| *PAO1 lasR-lux* | *lasR-luxCDABE* genomic reporter fusion in PAO1, Tc resistant | (18) |

[a]Tc, tetracycline.

## HPLC-Q-TOF-MS/MS analysis

Chromatographic separation of n-BuOH extract was performed with a Shimadzu Prominence high-performance liquid chromatography (HPLC) system using a COSMOSIL 5C18-MS-II Column (4.6 × 150 mm, 5 µm). The mobile phase consisted of two solvents: 0.1% aqueous formic acid (A) and 0.1% aqueous formic acid acetonitrile (B). The gradient program was optimized as follows: 0%–5% B at 0–10 min, 5%–50% B at 2–25 min, 50%–95% B at 25–35 min, 95% B at 35–37 min, and 95%–5% B at 37–38 min. The flow rate was set at 1 mL/min, and an aliquot of 5 µL was set as the injection volume.

Mass spectrometric determination was carried out on an X500R QTOF mass spectrometer with the ESI source in negative mode. The conditions were optimized as follows: ion spray voltage: −4,500 V, ion source temperature: 550°C, gas 1 and 2, nitrogen, 55 psi, and curtain gas, nitrogen, 35 psi. The collision energy (CE) was set at −10 V/−35 V and the declustering potential (DP) was −80 V. For the full MS-IDA (information dependent acquisition) analysis, the MS data were produced across the mass range of m/z from 100 to 1500 Da.

## HPLC analysis

Samples were dissolved in methanol by ultrasound and were filtered through 0.22-µm membrane filters before analysis. The quantitative analysis of gallic acid, protocatechuic acid, bergenin, and chlorogenic acid was measured by an Agilent 1200 liquid chromatography system (Agilent Technologies, USA) equipped with a degasser, quaternary pump, an auto-sampler, and a DAD detector. An Eclipse XDB -C18 Column (5 µm, 4.6 mm × 150 mm, Agilent Technologies, USA), with a mobile phase consisting of solvent A (water containing 0.2% phosphoric acid) and solvent B (methanol), was used for separation. N-BuOH (10 µL) was injected at a 0.8 mL/min flow rate and detected at 254 nm.

## Statistical analysis

All experiments were performed independently in triplicate with pooled samples of biological replicates. Data were analyzed by one-way analysis of variance with a $P$-value of 0.05 indicating significance.

## RESULTS

### MIC determination

All plant extracts showed antibacterial activity against *P. aeruginosa* to varying degrees (MICs from 5 to 50 mg/mL) (Table 2). The n-BuOH extract showed the strongest antibacterial effect, with MICs of 10 and 5 mg/mL against PAO1 and ATCC 27853, respectively. The water and petroleum ether fractions showed the least activity against PAO1 and ATCC 27853. In this study, the 20% methanol used to dissolve the extracts had no antibacterial effect on the MIC assay. We selected the n-BuOH extract for further experiments due to its highest activity. The MICs for gallic acid, protocatechuic acid, bergenin, and chlorogenic acid were 2, 2, 6, and 4 mg/mL, respectively, against PAO1.

### N-BuOH extract differentially affects gene expression of virulence factors

We determined the maximum sub-inhibitory concentration of methanol and n-BuOH extract that did not affect the 24-h growth curve of PAO1 (data not shown). Using the extracts with the maximum sub-inhibitory concentrations, a luminescence assay was

**TABLE 2** MICs for individual extracts against PAO1 and ATCC 27853

| Strains | MIC (mg/mL) | | | |
| --- | --- | --- | --- | --- |
| | n-BuOH | Ethyl acetate | Water | Petroleum ether |
| PAO1 | 10 | 12.5 | 50 | 50 |
| ATCC 27853 | 5 | 6.25 | 25 | 25 |

performed to determine the effect of the plant extracts on the virulence-related gene expression of PAO1. In order to exclude the influence of solvent, the effect of methanol (2.5%) was also detected. We found that methanol (2.5%) can inhibit the expression of *lasI, lasR, rhlI, rhlR, phzA1, phzA2,* and *pilG* by 36%, 3%, 32%, 32%, 12%, 30%, and 18%, respectively (Fig. 1). N-BuOH extract (containing 2.5% methanol) can inhibit the expression of *lasI, lasR, rhlI, phzA1, phzA2,* and *pilG* by 49%, 33%, 58%, 33%, 73%, and 39%, respectively. Deducting the effect of solvent, the *lasI, lasR, rhlI, phzA1, phzA2,* and *pilG* were reduced by 0.87-, 0.70-, 0.74-, 0.79-, 0.57-, and 0.79-fold in the presence of the n-BuOH extract alone ($P < 0.05$).

## N-BuOH extract shows an inhibitory effect on biofilm formation and pyocyanin production

Methanol (2.5%) had no effect on biofilm formation and pyocyanin production (Fig. 2). N-BuOH extract at sub-inhibitory growth concentrations can inhibit biofilm formation by 46% compared with the control group. On days 3 and 7, the production of pyocyanin was reduced by 39% and 79%, respectively, with sub-inhibitory concentrations of n-BuOH extract (Fig. 3), which suggests that the n-BuOH extract can suppress the pathogenesis related to pyocyanin production.

## Flavonoids and phenolic acids are major components in n-BuOH extract of *S. stolonifera* Meeb

Eleven compounds were detected in n-BuOH extract. Six of them belonged to flavonoids, including quercetin and derivatives of quercetin (Table 3). Phenolic acids, such as norbergenin, bergenin, gallic acid, and protocatechuic acid, were also found.

It has been known that the main components of ethanol extracts from *S. stolonifera* Meeb contain bergenin, quercetin, protocatechuic acid, gallic acid, and β-sitosterol (21). Based on the HPLC-Q-TOF-MS/MS data, we therefore quantified the contents of gallic acid, protocatechuic acid, bergenin, chlorogenic acid, and quercetin (other chemicals listed in Table 3 were not determined by HPLC because standard samples were not commercially available). The total ion chromatogram (TIC) for the n-BuOH extract is

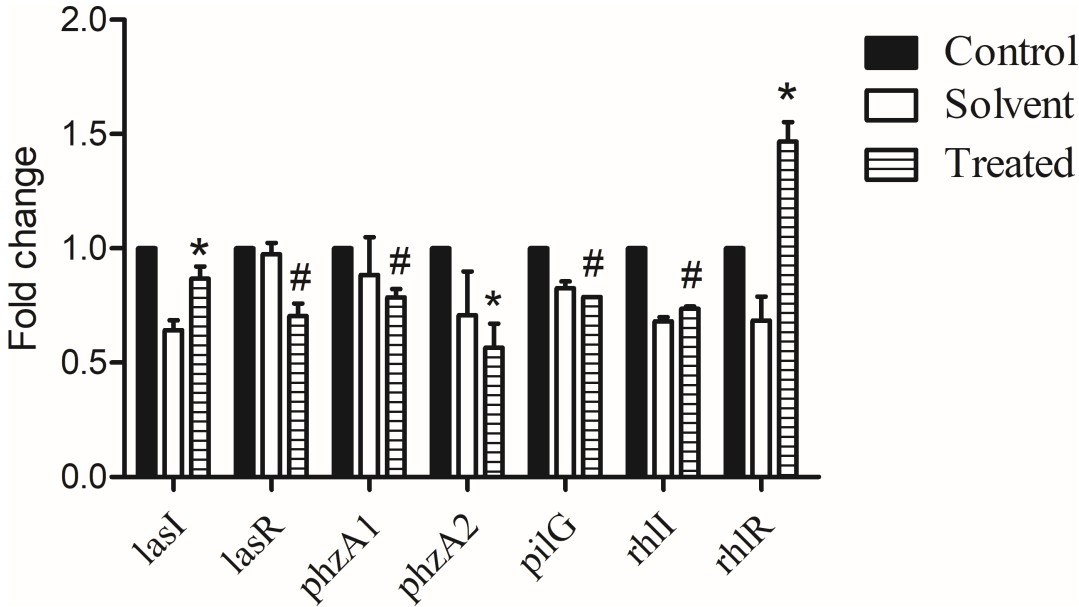

**FIG 1** Gene expression changes induced by n-BuOH extract of *S. stolonifera* Meeb, determined by luminescence assay. Relative fold change of maximum expression levels (cps) during the entire growth course of the PAO1 lux reporter strain in LB supplemented with methanol (solvent) alone (Control), or with sub-inhibitory concentrations of n-butyl alcohol extract (Treated). Control was set as 1, and the results are presented as the means ± SD of three independent experiments. *$P < 0.05$ versus the control group. #$P < 0.01$ versus the control group.

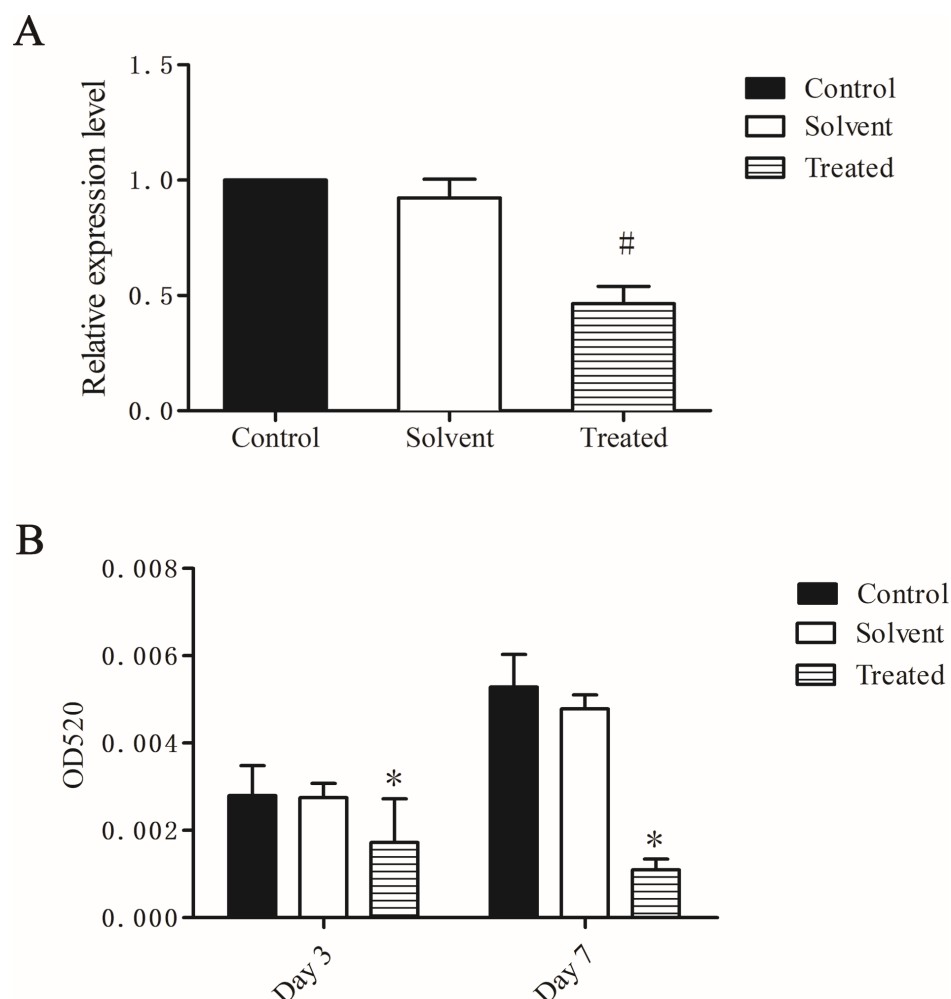

**FIG 2** Biofilm formation (A) and pyocyanin production (B) affected by n-BuOH extract of *S. stolonifera* Meeb. The results are presented as the means ± SD of three independent experiments. *$P < 0.05$ versus the control group. #$P < 0.01$ versus the control group.

illustrated in Fig. 3, with four compounds, i.e., gallic acid, protocatechuic acid, bergenin, and chlorogenic acid, being clearly identified (Fig. 4). Bergenin was the major chemical constituent in the dry extract, with a chemical content of 142.3 ± 0.2 mg/g, and the chemical contents of gallic acid, protocatechuic acid, and chlorogenic acid being 6.9 ± 0.2, 3.5 ± 0.1, and 3.2 ± 0.2 mg/g, respectively (Table 4). Quercetin was not detectable by HPLC, which suggests the low content of quercetin in the n-BuOH extract of *S. stolonifera* Meeb.

## Effects of gallic acid, protocatechuic acid, bergenin, and chlorogenic acid on biofilm formation, pyocyanin production, and virulence factor gene expression

The effects of the four identified compounds at maximum SIC on biofilm formation and pyocyanin production are shown in Fig. 5. Protocatechuic acid, gallic acid, and chlorogenic acid reduced biofilm formation by 40%, 35% and 16% ($P < 0.05$). Gallic acid, protocatechuic acid, and chlorogenic acid reduced the production of pyocyanin by 22%, 20%, and 50% ($P < 0.05$). Bergenin had no effect on biofilm formation and promoted the production of pyocyanin.

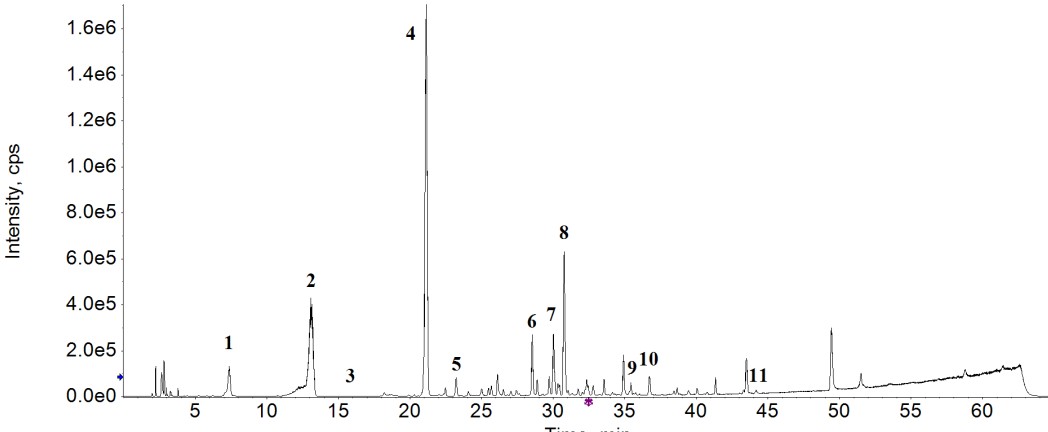

IDA Survey from HEC-LNP-0305.wiff2 (sample 1) - HEC-LNP-0305, Experiment 1, -IDA TOF MS (100 - 1500)

**FIG 3** The total ion chromatogram (TIC) for the n-BuOH extract of *S. stolonifera* (L.) Meeb. gallic acid (peak 1), norbergenin (peak 2), protocatechuic acid (peak 3), bengenin (peak 4), chlorogenic acid (peak 5), quercetin 3-O-gentiobioside (peak 6), quercetin-3-O-β-D-xylopyranosyl-(1→2)-β-D-galactopyranoside (peak 7), quercetin 3′-O- β-D-galactopyranoside (peak 8), quercetin 3-O- β-D-rhamnoside (peak 9), diquertin (peak 10), and quercetin (peak 11).

The effects of the four identified compounds at maximum SIC on gene expression of virulence factors are shown in Fig. 6. Gallic acid inhibited the expression of *lasI*, *phzA1*, *phzA2*, and *rhlR* by 37%, 33%, 9%, and 22%, respectively ($P < 0.05$). Protocatechuic acid inhibited the expression of *lasI*, *lasR*, *phzA1*, *phzA2*, *pilG*, *rhlI* by 32%, 41%, 12%, 8%, 15%, and 25%, respectively ($P < 0.05$). Chlorogenic acid inhibited the expression of *lasI*, *lasR*, *phzA1*, *phzA2*, *pilG*, *rhlI*, and *rhlR* by 78%, 42%, 80%, 55%, 33%, 79%, and 27%, respectively ($P < 0.05$). To our surprise, bergenin increased the expression of *phzA1* and *phzA2* by 60% and 18%, respectively ($P < 0.05$).

## DISCUSSION

The final products of the genes characterized in this study are important virulence factors that are involved in quorum sensing, motility, and bacteria-host interactions. The QS system regulates the formation of biofilm and the expression of many virulence factors in *P. aeruginosa* (22).

In order to determine whether the decreased expression of the tested genes results in the phenotypic changes of the PAO1 strain, we characterized the effects of the n-BuOH extract on production of pyocyanin and biofilm formation. Most of the antibiotics used to treat *P. aeruginosa* infections must cross a biofilm, which represents a protective mode of growth, since biofilm as a barrier blocks the ability of a variety of antibiotics to reach the bacteria, causing multi-drug resistance (23, 24). Biofilm formation is also

**TABLE 3** Qualitative analysis of chemical constituents of the *n*-BuOH extract of *S. stolonifera* (L.) Meeb by HPLC-Q-TOF-MS/MS

| | $t_R$/min | Elemental compositions | [M-H]$^-$ | Major fragment ion (*m/z*) | Tentative identification |
|---|---|---|---|---|---|
| 1 | 7.377 | $C_7H_6O_5$ | 169.0137 | 169, 125, 79, 69 | Gallic acid |
| 2 | 13.089 | $C_{13}H_{14}O_9$ | 313.0556 | 313, 235, 193, 165, 137 | Norbergenin |
| 3 | 15.690 | $C_7H_6O_4$ | 153.0195 | 153, 109, 91, 81 | Protocatechuic acid |
| 4 | 21.149 | $C_{14}H_{16}O_9$ | 327.0714 | 327, 312, 234, 192 | Bergenin |
| 5 | 23.235 | $C_{16}H_{18}O_9$ | 353.0874 | 191 | Chlorogenic acid |
| 6 | 28.570 | $C_{27}H_{30}O_{17}$ | 625.1410 | 625, 463, 300 | Quercetin 3-O-gentiobioside |
| 7 | 30.019 | $C_{26}H_{28}O_{16}$ | 595.1291 | 595, 300 | Quercetin-3-*O*-β-*D*-xylopyranosyl-(1→2)-*β*-*D*-galactopyranoside |
| 8 | 30.765 | $C_{21}H_{20}O_{12}$ | 463.0876 | 463, 301, 151 | Quercetin 3′-*O*-β-D-galactopyranoside |
| 9 | 35.471 | $C_{21}H_{20}O_{11}$ | 447.0938 | 447, 300, 271, 151 | Quercetin 3-O-β-D-rhamnoside |
| 10 | 36.736 | $C_{15}H_{12}O_7$ | 303.0503 | 303, 151, 123 | Diquertin |
| 11 | 44.187 | $C_{15}H_{10}O_7$ | 301.0349 | 301, 178, 151, 121, 107 | Quercetin |

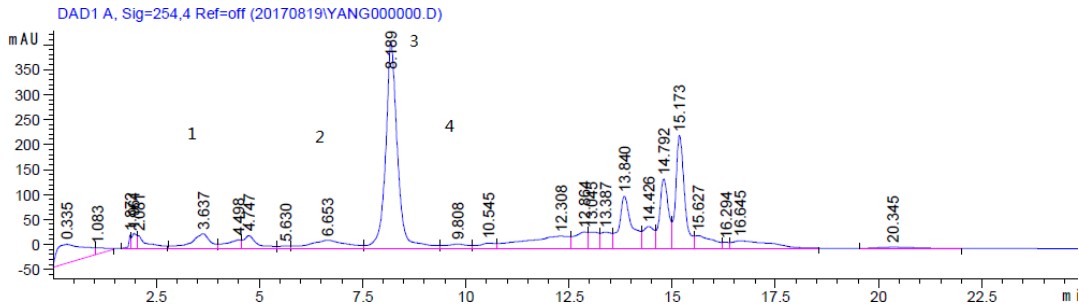

**FIG 4** Chromatograms of the identified compounds in mixed standard solutions, gallic acid (peak 1), protocatechuic acid (peak 2), bergenin (peak 3), and chlorogenic acid (peak 4)

closely correlated with infection and is involved in establishing chronic infections that are partly regulated by both the *las* and *rhl* systems (25, 26). Therefore, inhibiting biofilm formation or the ability to remove a biofilm would be helpful for the treatment of *P. aeruginosa* infections. In the present study, the n-BuOH extract of *Saxifraga stolonifera* Meeb suppressed the expression of the *lasI, lasR,* and *rhlI* genes, whereas *rhlR* was unchanged. The *las* and *rhl* systems are important components of the QS network and co-regulate the biofilm formation of *P. aeruginosa* (9). The *las* system plays an important role in biofilm formation because it is considered to be the top of the QS network (27). *LasR,* as the transcriptional regulator of the *las* system, not only regulates the *las* system, but also influences the expression of the *rhl* and *pqs* systems (28). The *rhlI* gene favors biofilm formation in *P. aeruginosa* by enhancing the production of matrix exopolysaccharide through the *pel* operon, and biofilm formation in *lasI* and *rhlI* mutants is severely reduced (29). In our luminescence assay experiments, we also showed that the expression of *pilG* is suppressed by the n-BuOH extract. *PilG* is one of the factors that influences the twitching motility of *P. aeruginosa* (30). Twitching motility is required for biofilm development (31). Thus, the downregulation of *pilG, lasR, lasI,* and *rhlI* caused by our n-BuOH extract may be the reason for the reduced biofilm formation.

Pyocyanin is one of the major virulence factors in *P. aeruginosa* and contributes to both acute and chronic infection (32, 33). It is also one of the major factors that suppresses lymphocyte proliferation. Pyocyanin can also exacerbate *P. aeruginosa* infections by promoting the formation of biofilms and delaying recovery of burn wounds (34). Pyocyanin production is regulated by the QS system. *RhlR* upregulates the synthesis of pyocyanin and rhamnolipid (35). The two homologous operons, *phzA1* and *phzA2,* are involved in the synthesis of phenazine, a pyocyanin precursor (34). We show that when *P. aeruginosa* grows in the presence of 2.5 mg/mL n-BuOH extract, the production of pyocyanin is reduced by 0.40- and 0.89-fold on the 3rd and 7th days in Fig. 3, which suggests that the n-BuOH extract can suppress the pathogenesis related to pyocyanin production, and the expression of *phzA1* and *phzA2* is both inhibited. Therefore, it is possible that the reduced production of pyocyanin might be due to inhibition of *rhlR, phzA1,* and *phzA2* expression.

Flavonoids and phenolics in n-BuOH extract are responsible for both antibacterial and anti-virulence effects. Although bergenin accounted for more than 14% of the total content of n-BuOH extract, in this study, bergenin alone had no antibacterial or anti-QS effects against standard *P. aeruginosa* strains at concentrations up to 3 mg/mL.

**TABLE 4** Phytochemical constituents in n-BuOH extract quantified by HPLC

| Analyte | Rt (min) | Concentration (mg/g) |
| --- | --- | --- |
| Gallic acid | 3.7 | 6.9 ± 0.2 |
| Protocatechuic acid | 6.4 | 3.5 ± 0.1 |
| Bergenin | 8.3 | 142.3 ± 0.2 |
| Chlorogenic acid | 9.8 | 3.2 ± 0.2 |

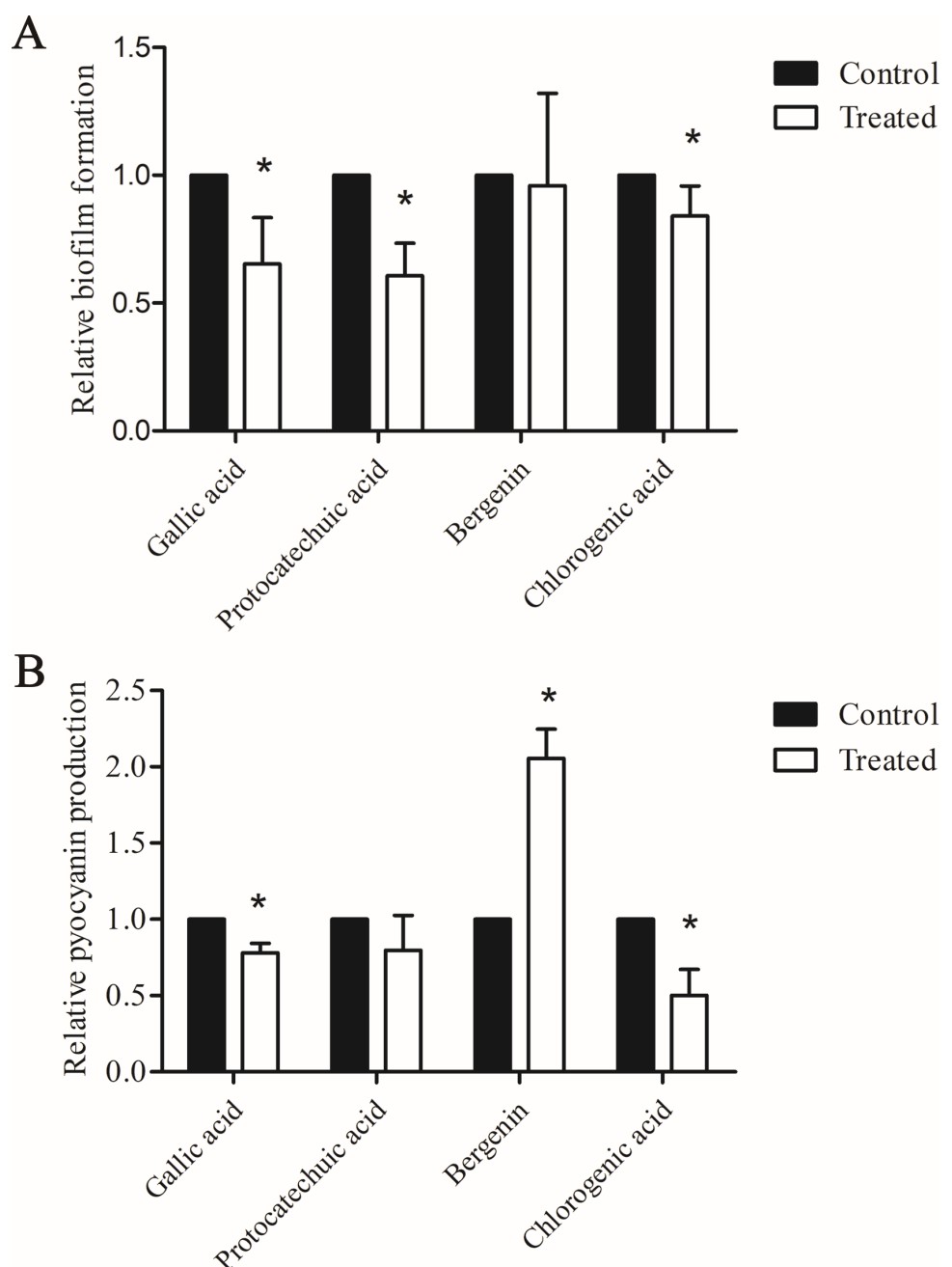

**FIG 5** Biofilm formation (A) and pyocyanin production (B) affected by gallic acid, protocatechuic acid, bergenin, and chlorogenic acid. The results are presented as the means ± SD of three independent experiments. *$P < 0.05$ versus the control group.

Flavonoids from multiple plants have been reported to be effective in inhibiting the growth of *P. aeruginosa* (36, 37). Galvão et al. proposed that the antibacterial mechanism of flavonoids is related to their binding to the *P. aeruginosa* cell wall (36). As the n-BuOH extract is rich in flavonoids, we therefore consider that the flavonoids contribute to the anti-*P. aeruginosa* effect of n-BuOH extract from *S. stolonifera* Meeb. Flavonoids (including quercetin, baicalein, naringenin, and kaempferol) and flavonoid-rich fractions from some herbs have also been reported to have anti-quorum sensing and anti-virulence effects against *P. aeruginosa* (38–40). Paczkowski et al. showed that flavonoids inhibit *P. aeruginosa* QS systems through antagonism *lasR* and *rhlR* (41). These studies

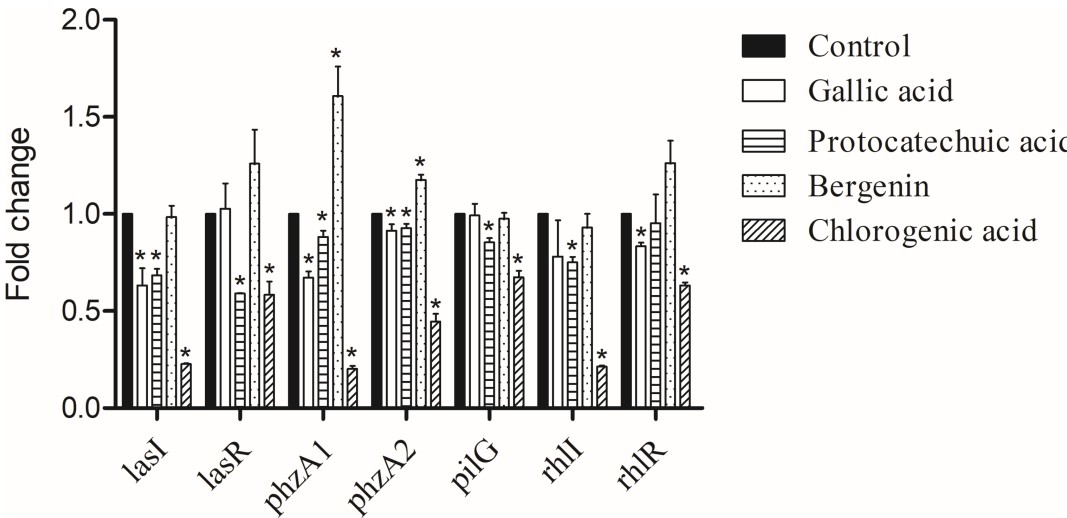

**FIG 6** Effects of gallic acid, protocatechuic acid, bergenin, and chlorogenic acid on the indicated virulence factor gene expression. *P < 0.05 versus the control group.

suggest that flavonoids might also play key roles in the anti-quorum sensing effect of n-BuOH extract from *S. stolonifera* Meeb.

Another major group of compounds from *Saxifraga stolonifera* Meeb is phenolic acids. Several studies demonstrate that phenolic acids are responsible for the antibacterial activity of some plants against *P. aeruginosa* (42, 43). The mechanism of the anti-*P. aeruginosa* effect of phenolic acids is related to their ability to damage the cell wall and cytoplasmic membrane (44). It may play an important role in inhibiting the virulence expression of foodborne bacteria and their prevention and control (45). Phenolic acids can inhibit the QS system of *P. aeruginosa* isolates, thereby influencing the expression of virulence factors, for example, by reducing pyocyanin production and inhibiting biofilm formation and swarming motility (28, 42). Thus, phenolic acids might partly explain the antibacterial and anti-virulence effects of n-BuOH extract from *S. stolonifera* Meeb.

## Conclusion

Our results demonstrate that n-BuOH extract of *S. stolonifera* Meeb has both antibacterial and anti-virulence effects on *P. aeruginosa*. N-BuOH extracts of *S. stolonifera* Meeb may be utilized in the future as an antibacterial and anti-virulence agent for the treatment of *P. aeruginosa*.

### ACKNOWLEDGMENTS

The authors wish to thank Dr. Kangmin Duan for gifts of strains used in this study and Dr. Stanley Lin for the English language editing. This work was financially supported by National Natural Science Foundation of China (81273562 and 81760664) and Natural Science Foundation of Guangdong Province (2018A0303130159).

Conceived and designed the experiments: F.Y. and Q.P. Performed the experiments: W.C., Z.Z., L.L., Y. Huang, L.C., Y.Z., Y.L., Y. Hong, and Q.H. Analyzed the data and initially wrote the manuscript: W.C., Z.Z., Q.P., and F.Y. supervised the study and prepared the manuscript. All authors read and approved the final manuscript.

### AUTHOR AFFILIATIONS

[1]Department of Pharmacy, The First Affiliated Hospital of Shantou University Medical College, Shantou, Guangdong, China
[2]Department of Pharmacology, Shantou University Medical College, Shantou, Guangdong, China

[3]Department of Clinical Laboratory, The First Affiliated Hospital of Shantou University Medical College, Shantou, Guangdong, China

[4]Molecular Microbiology Laboratory, Key Laboratory of Resources Biology and Biotechnology in Western China, Ministry of Education, Faculty of Life Sciences, Northwest University, Shanxi, China

[5]Department of Research and Education, The First Affiliated Hospital of Shantou University Medical College, Shantou, Guangdong, China

[6]Pediatrics, The Third People's Hospital of Datong, Datong, Shanxi, China

[7]The Ruth and Bruce Rappaport Faculty of Medicine, Technion-Israel Institute of Technology, Haifa, Israel

[8]Department of Pharmacy, The Second Affiliated Hospital of Shantou University Medical College, Shantou, Guangdong, China

[9]Central Laboratory of The Second Affiliated Hospital, School of Medicine, The Chinese University of Hong Kong, Shenzhen & Longgang District People's Hospital of Shenzhen, Shenzhen, Guangdong, China

## AUTHOR ORCIDs

Weidong Chen http://orcid.org/0000-0002-8228-331X
Lin Chen http://orcid.org/0000-0003-2322-4595
Qing Peng http://orcid.org/0000-0002-7790-6455
Fen Yao http://orcid.org/0000-0002-4959-0916

## FUNDING

| Funder | Grant(s) | Author(s) |
| --- | --- | --- |
| National Natural Science Foundation of China | 81273562 | Fen Yao |
| Natural Science Foundation of Guangdong Province | 2018A0303130159 | Fen Yao |

## AUTHOR CONTRIBUTIONS

Weidong Chen, Methodology, Writing – original draft | Zijie Zhang, Methodology, Writing – original draft | Yuanchun Huang, Methodology | Lin Chen, Methodology | Yijing Zhuang, Methodology | Yue Li, Methodology | Yuxiang Hong, Methodology | Lei Liu, Methodology | Qin He, Validation | Qing Peng, Conceptualization, Project administration, Writing – review and editing | Fen Yao, Conceptualization, Funding acquisition, Writing – review and editing

## ADDITIONAL FILES

The following material is available online.

Open Peer Review

**PEER REVIEW HISTORY (review-history.pdf).** An accounting of the reviewer comments and feedback.

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
