## [Reviewer comments · Microbiology Spectrum]

Microbiology Spectrum

Antibacterial and anti-virulence effects of *Saxifraga stolonifera* Meeb extracts against *Pseudomonas aeruginosa*

Fen Yao, weidong chen, zijie zhang, yue li, yuxiang hong, yuanchun huang, Lin Chen, yijing zhuang, lei liu, Qin He, and qing peng

Corresponding Author(s): Fen Yao, Shantou University Medical College

Review Timeline:

Submission Date:	September 11, 2025
Editorial Decision:	October 22, 2025
Revision Received:	December 13, 2025
Accepted:	January 5, 2026

Editor: Krisztina Papp-Wallace

Reviewer(s): Disclosure of reviewer identity is with reference to reviewer comments included in decision letter(s). The following individuals involved in review of your submission have agreed to reveal their identity: Iqbal Ahmad (Reviewer #1); Biao Tang (Reviewer #2)

Transaction Report:

DOI: <https://doi.org/10.1128/spectrum.02821-25>

Re: Spectrum02821-25 (Antibacterial and anti-virulence effects of *Saxifraga stolonifera* Meeb extracts against *Pseudomonas aeruginosa*)

Dear Prof. Fen Yao:

Thank you for the privilege of reviewing your work. Below you will find my comments, instructions from the Spectrum editorial office, and the reviewer comments.

Revision Guidelines

Sincerely,
Krisztina Papp-Wallace
Editor
Microbiology Spectrum

Reviewer #1 (Comments for the Author):

The authors have made a good effort to explore the Antibacterial and anti-virulence effects of *Saxifraga stolonifera* Meeb extracts against *Pseudomonas aeruginosa*. The chemistry part is well described. The active fraction showed the presence of 11 bioactive compounds with demonstrated activity. The methodological part of the gene expression-based study should have been described in more detail.

The study provided a good basis to explore and exploit the bioactive plant extract under study. However, in vivo efficacy of such bioactive extract/compounds needs to be evaluated.

The authors have not made any efforts to identify the active compounds through TLC bioautography. The HPTLC chromatogram should have been generated on the bioactive extract.

Certain minor corrections/comments arise on the manuscript as follows.

- ATCC27853: First time use, write the bacterial name with ATCC number for this strain
 - Minimum inhibitory concentration (MIC) values of 10 mg/ml and 5 mg/ml, respectively, are very low in potency as an antibacterial activity. However, inhibition of QS-linked property and gene expression at sub-MIC is interesting.
 - 1/4 MIC concentration was used. Why was the 1/2 MIC concentration not tested? Was the log CFU count of test strains determined with and without treatment? Please provide the growth kinetic data.
 - The MICs for gallic acid, protocatechuic acid, bergenin, and chlorogenic acid were determined and found to be 2, 2, 6 and 4 mg/ml, respectively, against PAO1. Are these compounds isolated and characterised, or were pure compounds purchased?
 - *Pseudomonas Aeruginosa*: please correct the spelling throughout the manuscript. *Pseudomonas aeruginosa*
- The authors are advised to revise the manuscript carefully, including corrections in the reference section and clearly address all issues.

Reviewer #2 (Comments for the Author):

Pseudomonas aeruginosa exhibits antibiotic resistance mainly through biofilm formation, and its virulence factor pyocyanin plays a key role in quorum sensing and the regulation of virulence gene expression. This study found that the n-BuOH extract of *Saxifraga stolonifera* exerts inhibitory effects on *P. aeruginosa* by suppressing quorum sensing, biofilm formation, motility, and the expression of virulence-related genes. Further investigation revealed that phenolic acid compounds in the extract play a major role; for instance, protocatechuic acid inhibits biofilm formation, and chlorogenic acid reduces pyocyanin production and downregulates virulence gene expression. These findings provide a theoretical basis and potential application for using *S. stolonifera* extracts as herbal feed additives or in combination with antibiotics to combat infections.

Comments and Suggestions:

1. Please provide photos from MIC and crystal violet assays to visually demonstrate the inhibitory effects of the n-BuOH extract, as well as to show that the water, petroleum ether, and 20% methanol extracts are less effective. Similarly, provide MIC and crystal violet photos for the four phenolic acid compounds.

2. Line 209 mentions that the effects of methanol and the n- n-BuOH extract on PAO1 growth curves were not shown. It is recommended to include growth curves under various concentrations of methanol and n-BuOH extract, not only at SIC levels.

3. The use of luminescent reporter strains to measure virulence gene expression is a major strength of the paper. Please briefly describe the principle in the Methods section. Additionally, consider verifying virulence gene expression by qPCR, since the insertion of the lux cassette may influence target gene expression.

4. In Result 2, the authors note that methanol solvent itself affects virulence gene expression, and the solvent effect was subtracted when analyzing the n-BuOH extract. Please clarify how this correction was performed during data processing. Also, what is the difference between the Control and Solvent groups? Was the Treated group prepared from *S. stolonifera* n-BuOH extract concentrated under 50 {degree sign}C and redissolved in methanol?

5. Line 264: Does the n-BuOH extract show inhibitory effects on other pathogens such as *Salmonella* or pathogenic *E. coli*? Please expand the discussion and cite relevant literature (e.g., DOI: 10.1016/j.ijfoodmicro.2023.110120; DOI: 10.1128/mbio.00651-25; DOI: 10.1128/mSphere.00125-21) .

6. The current figures appear to be of relatively low quality. Please enhance the resolution and overall presentation. Moreover, incorporating additional quantitative data would strengthen the validity and reliability of the conclusions.

7. Minor text and formatting issues:

a) Line 36: "lasA" does not appear elsewhere in the text - please confirm.

b) Line 123: "Mueller-Hinton broth (MH) broth" → "Mueller-Hinton (MH) broth".

c) Line 147: "each well. at room temperature" → "each well at room temperature".

d) Line 187: "N- BuOH" → "N-BuOH"; "10µl" → "10 µL".

e) Line 283: "LasRi" should likely be "LasR".

Subject: Spectrum02821-25 Response to Reviewers

Re: Spectrum02821-25 (Antibacterial and anti-virulence effects of *Saxifraga stolonifera* Meeb extracts against *Pseudomonas aeruginosa*)

Dear editor:

We would like to express our sincere gratitude for the valuable revision suggestions you and the Reviewers provided on this manuscript. These suggestions are of great help in improving the quality of the paper, and we have carefully revised the manuscript in accordance with all the suggestions.

Additionally, as some supplementary experiments have been added, an author, Qin He, has been included in this manuscript. Her email address is: h1887977@163.com.

The responses and revisions to the reviewers' comments are as follows:

Response to Reviewer #1:

The authors have made a good effort to explore the Antibacterial and anti-virulence effects of *Saxifraga stolonifera* Meeb extracts against *Pseudomonas aeruginosa*. The chemistry part is well described. The active fraction showed the presence of 11 bioactive compounds with demonstrated activity. The methodological part of the gene expression-based study should have been described in more detail.

The study provided a good basis to explore and exploit the bioactive plant extract under study. However, in vivo efficacy of such bioactive extract/compounds needs to be evaluated.

The authors have not made any efforts to identify the active compounds through TLC bioautography. The HPTLC chromatogram should have been generated on the bioactive extract.

Author Response and Revisions: We have added the principle of the lux-based reporters in the Methods section, revised "Using the lux-based reporters, gene expression in liquid cultures was determined as counts per second (cps) of light production." in line 128 to "The expression of genes was determined using the lux-based reporters: the promoter regions of the gene were cloned into the upstream of the lux genes on plasmid pMS402, gene expression in liquid cultures was determined as counts per second (cps) of light production.", and included additional literature citations (reference 16).

We regret that we do not have the conditions to conduct in vivo experiments. Since the previous *Saxifraga stolonifera* extract has been used up, new *Saxifraga stolonifera* cannot be purchased in a short period of time, and there is not enough personnel to carry out the extraction work, we are also unable to supplement the TLC bioautography experiment.

Comment 1: ATCC27853: First time use, write the bacterial name with ATCC number for this strain.

Author Response and Revisions: "ATCC27853" was first used in the Abstract, specifically in the line 33. We have revised the original description of "*P. aeruginosa* (PAO1) and ATCC27853" to "*P. aeruginosa* PAO1 and *P. aeruginosa* ATCC27853". Additionally, in the original version of the manuscript, within the "Bacterial strains and media" section of "Materials and methods", there was also an introduction to ATCC27853 in Table 1.

Comment 2: Minimum inhibitory concentration (MIC) values of 10 mg/ml and 5 mg/ml, respectively, are very low in potency as an antibacterial activity. However, inhibition of QS-linked property and gene expression at sub-MIC is interesting.

Author Response and Revisions: The potency of n-BuOH extract in antibacterial activity is not high; therefore, we focus on discussing its role in the virulence and QS system of *P. aeruginosa*.

Comment 3: 1/4 MIC concentration was used. Why was the 1/2 MIC concentration not tested? Was the log CFU count of test strains determined with and without treatment? Please provide the growth kinetic data.

Author Response and Revisions: We determined the effect of n-BuOH extract and four phenolic acid compounds on the bacterial growth curve. The n-BuOH extract affects the growth of PAO1 at the concentration of 1/2 MIC, so we used the concentration of 1/4 MIC instead. The results of the growth curve are presented in Figure S1 of the newly added "Miscellaneous File".

Comment 4: The MICs for gallic acid, protocatechuic acid, bergenin, and chlorogenic acid were determined and found to be 2, 2, 6 and 4 mg/ml, respectively, against PAO1. Are these compounds isolated and characterised, or were pure compounds purchased?

Author Response and Revisions: These compounds are pure products purchased directly.

Comment 5: *Pseudomonas Aeruginosa*: please correct the spelling throughout the manuscript. *Pseudomonas aeruginosa*. The authors are advised to revise the manuscript carefully, including corrections in the reference section and clearly address all issues.

Author Response and Revisions: We have revised all instances of "*Pseudomonas Aeruginosa*" to "*Pseudomonas aeruginosa*", with revisions primarily focused on the "References" section. Due to the use of an incorrect EndNote output style, the format has now been updated to the ASM style.

Response to Reviewer #2:

Pseudomonas aeruginosa exhibits antibiotic resistance mainly through biofilm formation, and its virulence factor pyocyanin plays a key role in quorum sensing and the regulation of virulence gene expression. This study found that the n-BuOH extract of *Saxifraga stolonifera* exerts inhibitory effects on *P. aeruginosa* by suppressing quorum sensing, biofilm formation, motility, and the expression of virulence-related genes. Further investigation revealed that phenolic acid compounds in the extract play a major role; for instance, protocatechuic acid

inhibits biofilm formation, and chlorogenic acid reduces pyocyanin production and downregulates virulence gene expression. These findings provide a theoretical basis and potential application for using *S. stolonifera* extracts as herbal feed additives or in combination with antibiotics to combat infections.

Comments and Suggestions:

Comment 1: Please provide photos from MIC and crystal violet assays to visually demonstrate the inhibitory effects of the n-BuOH extract, as well as to show that the water, petroleum ether, and 20% methanol extracts are less effective. Similarly, provide MIC and crystal violet photos for the four phenolic acid compounds.

Author Response and Revisions: We sincerely apologize that we only retained the data of the MIC and the crystal violet assay previously, and did not keep the relevant photos.

We additionally provide the crystal violet photos of the four phenolic acid compounds in Figure S3 of the " Miscellaneous File ".

Comment 2: Line 209 mentions that the effects of methanol and the n-BuOH extract on PAO1 growth curves were not shown. It is recommended to include growth curves under various concentrations of methanol and n-BuOH extract, not only at SIC levels.

Author Response and Revisions: We investigated the effects of the n-BuOH extract at concentrations of 1/2 MIC, 1/4 MIC, and 1/8 MIC, as well as methanol at concentrations of 1.25%, 2.5%, 5%, and 10%, on the growth of PAO1. The results are presented in Figure S1 of the " Miscellaneous File ". The n-BuOH extract at 1/4 MIC and methanol at 2.5% had no impact on the growth of PAO1.

Comment 3: The use of luminescent reporter strains to measure virulence gene expression is a major strength of the paper. Please briefly describe the principle in the Methods section. Additionally, consider verifying virulence gene expression by qPCR, since the insertion of the lux cassette may influence target gene expression.

Author Response and Revisions: We have added the principle of the lux-based reporters in the Methods section, revised "Using the lux-based reporters, gene expression in liquid cultures was determined as counts per second (cps) of light production." in line 128 to "The expression of genes was determined using the lux-based reporters: the promoter regions of the gene were cloned into the upstream of the lux genes on plasmid pMS402, gene expression in liquid cultures was determined as counts per second (cps) of light production.", and included additional literature citations (reference 16).

We repurchased the four phenolic acid compounds to conduct the qPCR experiment. The results of the qPCR are provided in Figure S2 of the " Miscellaneous File ".

Overall, the qPCR results showed that under the action of gallic acid, protocatechuic acid, bergenin, and chlorogenic acid, The expression trends of the relevant virulence factor genes were basically consistent with those detected using luminescent reporter strains.

Comment 4: In Result 2, the authors note that methanol solvent itself affects virulence gene

expression, and the solvent effect was subtracted when analyzing the n-BuOH extract. Please clarify how this correction was performed during data processing. Also, what is the difference between the Control and Solvent groups? Was the Treated group prepared from *S. stolonifera* n-BuOH extract concentrated under 50 {degree sign}C and redissolved in methanol?

Author Response and Revisions: The Treated group was prepared from n-BuOH extract of *Saxifraga stolonifera* redissolved in methanol. The Solvent group contains the same concentration of methanol as that in the Treated group, while the Control group contains no methanol. During data processing, the effect of methanol was directly subtracted from the data of the Treated group. As *lasI*: methanol (2.5%) reduced the expression level of the *lasI* gene by 36%, while the Treated group reduced by 49%. After deducting the effect of methanol, it can be concluded that the n-BuOH extract alone reduces the expression level of the *lasI* gene by 13%. To eliminate ambiguity, we have revised "Deducting the effect of solvent, the *lasI*, *lasR*, *rhlI*, *phzA1*, *phzA2* and *pilG* were reduced by 0.87-fold, 0.70-fold, 0.74-fold, 0.79-fold, 0.57-fold, and 0.79-fold in the presence of the n-BuOH extract (P<0.05). "in Line 215 of the original manuscript to " n-BuOH extract (containing 2.5% methanol) can inhibit the expression of *lasI*, *lasR*, *rhlI*, *phzA1*, *phzA2* and *pilG* by 49%, 33%, 58%, 33%, 73%, 39%. Deducting the effect of solvent, the *lasI*, *lasR*, *rhlI*, *phzA1*, *phzA2* and *pilG* were reduced by 0.87-fold, 0.70-fold, 0.74-fold, 0.79-fold, 0.57-fold, and 0.79-fold in the presence of the n-BuOH extract alone (P<0.05).".

Comment 5: Line 264: Does the n-BuOH extract show inhibitory effects on other pathogens such as Salmonella or pathogenic E. coli? Please expand the discussion and cite relevant literature (e.g., DOI: 10.1016/j.ijfoodmicro.2023.110120; DOI: 10.1128/mbio.00651-25; DOI: 10.1128/mSphere.00125-21).

Author Response and Revisions: We have supplemented the introduction and discussion sections of the paper with content regarding the significance of *Saxifraga stolonifera* extracts as potential herbal feed additives in preventing the development of drug resistance in the animal husbandry industry, with relevant literature cited. The specific content can be found in lines 83 and lines 331.

Comment 6: The current figures appear to be of relatively low quality. Please enhance the resolution and overall presentation. Moreover, incorporating additional quantitative data would strengthen the validity and reliability of the conclusions.

Author Response and Revisions: We have re-exported the figures at a resolution of 600 dpi using GraphPad Prism. Additionally, we have supplemented other required quantitative data, which has been submitted in the " Miscellaneous File ".

Comment 7: Minor text and formatting issues:

- a) Line 36: "*lasA*" does not appear elsewhere in the text - please confirm.
- b) Line 123: "Mueller-Hinton broth (MH) broth" → "Mueller-Hinton (MH) broth".
- c) Line 147: "each well. at room temperature" → "each well at room temperature".

d) Line 187: "N- BuOH" → "N-BuOH"; "10μl" → "10 μL".

e) Line 283: "*LasRi*" should likely be "*LasR*".

Author Response and Revisions: Thank you for pointing out the minor text and formatting issues. The "*lasA*" in Line 36 is a typo, and all other formatting issues have been corrected.

Re: Spectrum02821-25R1 (Antibacterial and anti-virulence effects of *Saxifraga stolonifera* Meeb extracts against *Pseudomonas aeruginosa*)

Dear Prof. Fen Yao:

Your manuscript has been accepted, and I am forwarding it to the ASM production staff for publication. Your paper will first be checked to make sure all elements meet the technical requirements. ASM staff will contact you if anything needs to be revised before copyediting and production can begin. Otherwise, you will be notified when your proofs are ready to be viewed.

Sincerely,
Krisztina Papp-Wallace
Editor
Microbiology Spectrum